# Pawsterior: Variational Flow Matching for Structured Simulation-Based Inference

**Jorge Carrasco-Pollo**[*]
University of Amsterdam
`jorge.carrasco.pollo@student.uva.nl`

**Floor Eijkelboom**[*]
UvA-Bosch Delta Lab
`f.eijkelboom@uva.nl`

**Jan-Willem van de Meent**
UvA-Bosch Delta Lab
`j.w.vandemeent@uva.nl`

## Abstract

We introduce *Pawsterior*, a variational flow-matching framework for improved and extended simulation-based inference (SBI). Many SBI problems involve posteriors constrained by structured domains—such as bounded physical parameters or hybrid discrete–continuous variables—yet standard flow-matching methods typically operate in unconstrained spaces. This mismatch leads to inefficient learning and difficulty respecting physical constraints. Our contributions are twofold. First, generalizing the geometric inductive bias of CatFlow, we formalize endpoint-induced affine geometric confinement, a principle that incorporates domain geometry directly into the inference process via a two-sided variational model. This formulation improves numerical stability during sampling and leads to consistently better posterior fidelity, as demonstrated by improved classifier two-sample test performance across standard SBI benchmarks. Second, and more importantly, our variational parameterization enables SBI tasks involving discrete latent structure (e.g., switching systems) that are fundamentally incompatible with conventional flow-matching approaches. By addressing both geometric constraints and discrete latent structure, Pawsterior provides a principled way to apply flow-matching in a broader range of structured SBI settings.

## 1 Introduction

Generative modeling enables learning complex distributions and sampling in high-dimensional domains such as images, molecules, and physical systems. Among recent approaches, *Flow Matching* (FM) (Lipman et al., 2023; Albergo & Vanden-Eijnden, 2023; Liu et al., 2023) has emerged as a flexible and scalable method for training continuous-time generative models by learning a velocity field that transports samples from a simple base distribution $p_0$ to a target distribution $p_1$. A central application in scientific and engineering workflows is *Simulation-Based Inference* (SBI) (Cranmer et al., 2020), where complex mechanistic simulators—such as climate models, biological systems, or particle physics simulations—define a stochastic mapping from parameters $\theta$ to observations $x$ while the likelihood remains intractable. The goal is to approximate the posterior $p(\theta \mid x)$ efficiently, making flow-based methods particularly attractive for amortized inference in these settings (Wildberger et al., 2023).

Despite this success, standard FM posterior estimators typically make a strong implicit assumption: they treat the parameter space as an unconstrained Euclidean vector space and learn a global vector field over the full ambient domain. In many SBI problems, however, posteriors exhibit *structured support* dictated by physical constraints, bounds, symmetries, or discrete latent structure. In particular, some posteriors do not live naturally in Euclidean space at all, but on constrained manifolds such as probability simplices, as in categorical or regime-switching models. Such structure is prevalent across applications including gravitational-wave inference (Ussipov et al., 2024; Magnall et al.,

---

[*]Equal contribution.

2025; Jin et al., 2026), biology (Vélez-Cruz & Papandreou-Suppappola, 2024), engineering (Kwao & Raptis, 2025), and sea-ice modeling (Finn et al., 2025). When probability mass concentrates on a feasible subset, unconstrained flows can traverse invalid regions—wasting capacity, violating constraints, and introducing spurious uncertainty—and in discrete settings this mismatch can lead to fundamental incompatibilities with conventional flow-matching formulations.

*Variational Flow Matching* (VFM) (Eijkelboom et al., 2024) offers a principled alternative to direct velocity regression by recasting FM as variational inference over the interpolation endpoints. Rather than regressing a velocity field in the ambient space, VFM learns a conditional distribution over the endpoint given the current state, making endpoint structure explicit and enabling constraints to be imposed at the level of the inferred posterior. This yields models that are both expressive and geometry-aware, and clarifies a key limitation of standard FM: under common parameterizations, FM can be understood as doing Gaussian variational inference over the data (mean-matching), which VFM generalizes and relaxes. Building on this perspective, VFM has been extended beyond Euclidean settings to a range of structured domains, including Riemannian geometries (Zaghen et al., 2025), molecular and graph-based generation (Eijkelboom et al., 2024; 2025), VQ image generation (Matişan et al., 2026), physical and biological systems (Sakalyan et al., 2025; Finn et al., 2025), and mixed or tabular data types (Guzmán-Cordero et al., 2025; Nasution et al., 2026).

In this work, we introduce *Pawsterior*[1], a variational flow-matching framework that resolves the mismatch between standard FM and the structured nature of simulation-based inference. Beyond improving posterior fidelity in conventional continuous-parameter SBI, Pawsterior extends flow-matching inference to settings that fall outside Euclidean assumptions, including constrained, discrete, and hybrid parameter spaces. Our contributions are twofold:

- We formalize *endpoint-induced affine geometric confinement* and develop a stable two-sided endpoint inference model that explicitly accounts for bounded and structured domains, improving posterior fidelity and numerical stability on standard SBI benchmarks.
- By shifting inference from Euclidean velocity regression to endpoint distributions, our variational formulation naturally supports categorical and mixed parameter spaces, enabling coherent amortized inference for problems—such as switching systems—that are fundamentally incompatible with conventional flow-matching approaches.

## 2 BACKGROUND

### 2.1 (VARIATIONAL) FLOW MATCHING

**Flow Matching.** Flow Matching (FM) (Lipman et al., 2023; Albergo & Vanden-Eijnden, 2023; Liu et al., 2023) learns a continuous-time *transport model* that maps a simple *prior* (noise) distribution $p_0$ to a target data distribution $p_1$. FM specifies an interpolation between endpoints $x_0 \sim p_0$ and $x_1 \sim p_1$, commonly the affine interpolation line

$$x_t := \alpha_t x_0 + \beta_t x_1, \tag{1}$$

which induces intermediate marginals $x_t \sim p_t$. It then learns a time-dependent velocity field $v_t^\varphi$ (parameterized by $\varphi$) whose flow satisfies

$$\frac{\mathrm{d}x_t}{\mathrm{d}t} = v_t^\varphi(x_t),$$

so that integrating from $t = 0$ transports $x_0 \sim p_0$ to a terminal state distributed as $p_1$ along the *probability path* $(p_t)_{t \in [0,1]}$.

A key point is that FM does *not* require the (generally intractable) marginal velocity field that exactly pushes $p_0$ to $p_1$. Instead, for a fixed interpolation, the target field is characterized by the conditional expectation of the time-derivative of the interpolation, e.g. in the straight-line case where we define $x_t := (1 - t)x_0 + t x_1$, we have that

$$v_t^\star(x_t) = \mathbb{E}[x_1 - x_0 \,|\, x_t], \tag{2}$$

and can be learned by conditional regression:

$$\mathcal{L}_{\mathrm{FM}}(\varphi) = \mathbb{E}_{t \sim \mathcal{U}[0,1]} \, \mathbb{E}_{x_0 \sim p_0,\, x_1 \sim p_1} \left[ \left\| v_t^\varphi(x_t) - (x_1 - x_0) \right\|_2^2 \right]. \tag{3}$$

[1]Our code implementation is available at https://github.com/Carrask0/pawsterior.

**Variational Flow Matching.**   Given that when $t < 1$ we have

$$x_t = tx_1 + (1-t)x_0 \iff x_0 = \frac{x_t - tx_1}{1-t}, \tag{4}$$

the marginal field can be written as an expectation over endpoint-conditioned fields

$$v_t^\star(x_t) = \mathbb{E}\left[u_t(x_t \mid x_1) \mid x_t\right] \quad \text{where} \quad u_t(x_t \mid x_1) := \frac{x_1 - x_t}{1-t}, \tag{5}$$

highlighting that FM implicitly depends on the (typically intractable) endpoint posterior $p_t(x_1 \mid x_t)$.

Variational Flow Matching (VFM; Eijkelboom et al. (2024)) makes this dependence explicit by replacing $p_t(x_1 \mid x_t)$ with a learned approximation $q_t^\varphi(x_1 \mid x_t)$, inducing the variationally parameterized velocity as follows:

$$v_t^\varphi(x_t) = \mathbb{E}_{q_t^\varphi(x_1 \mid x_t)}\left[u_t(x_t \mid x_1)\right]. \tag{6}$$

VFM fits $q_t^\varphi$ by minimizing $\mathrm{KL}\left(p_t(x_1, x_t) \mid\mid q_t^\varphi(x_1, x_t)\right)$, or, equivalently, maximizing the conditional log-likelihood of endpoints:

$$\mathcal{L}_{\mathrm{VFM}}(\varphi) = \mathbb{E}_t\left[\mathrm{KL}\left(p_t(x_1, x_t) \mid\mid q_t^\varphi(x_1, x_t)\right)\right] = \mathbb{E}_{t, x_0, x_1}\left[-\log q_t^\varphi(x_1 \mid x_t)\right] + \mathrm{const}, \tag{7}$$

where $t \sim \mathcal{U}[0, 1]$, $x_0 \sim p_0$, $x_1 \sim p_1$ and the constant is independent of $\varphi$.

In the affine case, the linearity of expectation implies that

$$\mathbb{E}_{p_t(x_1 \mid x_t)}\left[u_t(x_t \mid x_1)\right] = u_t\left(x_t \mid \mathbb{E}_{p_t(x_1 \mid x_t)}[x_1]\right), \tag{8}$$

and hence the expectation in Equation 6 depends only on the posterior mean. Therefore, in the affine case, a fully factorised variational distribution is not making an approximation: as long as it captures the correct posterior mean, it recovers the vector field exactly (Eijkelboom et al., 2024), i.e.

$$q_t^\varphi(x_1 \mid x_t) = \prod_{d=1}^D q_t^\varphi(x_1^d \mid x_t) \text{ such that } \mathcal{L}_{\mathrm{MF\text{-}VFM}}(\varphi) = -\mathbb{E}_{t, x_1, x_t}\left[\sum_{d=1}^D \log q_t^\varphi(x_1^d \mid x_t)\right]. \tag{9}$$

As such VFM naturally provides a scalable and distribution-aware FM approach. This recovers standard supervised losses per coordinate (Gaussian mean-matching for continuous variables; cross-entropy for categorical variables), enabling hybrid discrete–continuous endpoints.

Sampling still proceeds by integrating the induced flow. For the straight-line interpolation,

$$v_t^\varphi(x_t) = \frac{\mu_t^\varphi(x_t) - x_t}{1-t}, \tag{10}$$

where $\mu_t^\varphi(x_t) := \mathbb{E}_{q_t^\varphi}[x_1 \mid x_t]$, which is then integrated from $t = 0$ with $x_0 \sim p_0$ to obtain samples from $p_1$.

## 2.2   SIMULATION-BASED INFERENCE

In *simulation-based inference* (SBI), the goal is to infer underlying governing parameters $\theta$ from observed data $x$ where the data-generating process is defined by a simulator rather than an explicit likelihood function. Such problems are ubiquitous in scientific domains like physics, biology, and ecology, where high-fidelity simulators encode complex mechanistic knowledge—often involving stochastic dynamics or unobservable intermediate states.

From a Bayesian perspective, inference relies on the posterior distribution:

$$p(\theta \mid x) = \frac{p(x \mid \theta)\, p(\theta)}{p(x)}.$$

However, the core difficulty in the SBI setting is that the likelihood $p(x \mid \theta)$ is typically *intractable*. This intractability usually arises because the simulator generates data via a complex sequence of latent stochastic events; evaluating the likelihood would require marginalizing over all possible execution paths, which is computationally infeasible.

Crucially, while we cannot *evaluate* the likelihood density, we can *sample* from it by running the simulation:

$$x \sim \texttt{Simulator}(\theta) \iff x \sim p(x \mid \theta). \tag{11}$$

This capability enables likelihood-free inference by replacing analytic derivation with synthetic data generation.

A prominent class of SBI methods, *neural posterior estimation* (NPE), solves this inverse problem by approximating the posterior directly with a conditional density estimator $q^\varphi(\theta \mid x)$ (e.g., a normalizing flow). Training is performed using a dataset of parameters sampled from the prior, $\theta \sim p(\theta)$, and their corresponding simulated observations, $x \sim p(x \mid \theta)$. Once trained, the model enables *amortized inference*: posterior samples and density evaluations can be computed efficiently for any new observation $x$ without requiring further expensive simulations.

## 3 FLOW MATCHING ON STRUCTURED DOMAINS

### 3.1 MOTIVATION: ENDPOINT-INDUCED AFFINE CONFINEMENT

**Motivation.** Many simulation-based inference (SBI) problems have *structured* parameter spaces: parameters live on a feasible set $\Omega \subseteq \mathbb{R}^D$ determined by physical bounds, conservation laws, simplices, or hybrid discrete–continuous structure. Yet standard flow-matching posterior estimators are usually parameterized as unconstrained Euclidean vector fields on the full ambient space $\mathbb{R}^D$. In SBI, this mismatch is not just wasteful: it allocates capacity to directions that correspond to invalid simulator inputs and can push probability mass through regions where the simulator is undefined or unphysical.

Crucially, this mismatch is *not* inherent to FM at the population level, as seen e.g. in VFM for categorical data (*CatFlow*) (Eijkelboom et al., 2024). Even if the base distribution is unconstrained (e.g. Gaussian noise), the *population* FM target already "knows" the endpoint geometry. The reason is simple: the FM target is a *conditional* expectation given an intermediate state. Conditioning on $x_t$ restricts contributing endpoints to the feasible set $x_1 \in \Omega$, so the resulting average direction is automatically aligned with $\Omega$.

This suggests a design principle: instead of learning an unconstrained velocity field and hoping it discovers feasibility from data, we should parameterize the model so that this *endpoint-induced* confinement is explicit and therefore preserved under finite-sample training.

**Formalism.** As a representative setting, we assume that the data lives on a convex support $\Omega$.[2] Consider the affine interpolation

$$x_t = \alpha_t x_0 + \beta_t x_1, \tag{12}$$

where $x_0 \sim \mathcal{N}(0, I)$ is unconstrained and $x_1 \in \Omega$. Differentiating yields the instantaneous velocity along the interpolation,

$$v_t = \dot{\alpha}_t x_0 + \dot{\beta}_t x_1. \tag{13}$$

The probability-flow ODE uses the conditional expectation

$$u_t(x_t) := \mathbb{E}[v_t \mid x_t] = \dot{\alpha}_t \, \mathbb{E}[x_0 \mid x_t] + \dot{\beta}_t \, \mathbb{E}[x_1 \mid x_t]. \tag{14}$$

Assuming $\alpha_t \neq 0$, the interpolation identity gives

$$x_0 = \frac{x_t - \beta_t x_1}{\alpha_t} \quad \implies \quad \mathbb{E}[x_0 \mid x_t] = \frac{x_t - \beta_t \, \mathbb{E}[x_1 \mid x_t]}{\alpha_t}. \tag{15}$$

Substituting into equation 14 yields the affine decomposition

$$u_t(x_t) = a_t x_t + c_t \, \mathbb{E}[x_1 \mid x_t], \qquad a_t := \frac{\dot{\alpha}_t}{\alpha_t}, \qquad c_t := \dot{\beta}_t - a_t \beta_t, \tag{16}$$

where the scalar coefficients $(a_t, c_t)$ depend only on the interpolation schedule.

Since the conditional distribution of $x_1$ given $x_t$ is supported on $\Omega$, its conditional mean must also lie in $\Omega$:

$$\mathbb{E}[x_1 \mid x_t] \in \Omega. \tag{17}$$

Combining equation 16 and equation 17 gives the set inclusion

$$u_t(x_t) \in a_t x_t + c_t \, \Omega := \{a_t x_t + c_t y : y \in \Omega\}. \tag{18}$$

---

[2]In case of the discrete data, we consider the convex hull of the data support, i.e. the probability simplex.

We call equation 18 **endpoint-induced affine geometric confinement**: although noise samples live in the full ambient space, the *population* FM target at time $t$ lies in an affine image of the feasible set.

This observation motivates the constructions below. The geometry of $\Omega$ is already present in the population target, but standard Euclidean parameterizations need not respect it in finite-sample learning. We therefore seek an endpoint-based variational parameterization that (i) exposes endpoint structure explicitly and (ii) allows feasibility—and hence confinement—to be enforced by design.

## 3.2 Two-sided endpoint prediction for stable VFM

A direct way to exploit equation 16 is to predict only the constrained endpoint statistic $\mathbb{E}[x_1 \mid x_t]$ and recover $\mathbb{E}[x_0 \mid x_t]$ via equation 15. In practice, this one-sided recovery can be numerically fragile: common schedules induce large rescalings (e.g. division by $\alpha_t$ or $1 - t$), so small endpoint errors may be amplified into large velocity errors near the boundary of the time interval.

We therefore adopt a **two-sided** variational endpoint model that approximates the joint endpoint posterior,

$$p_t(x_0, x_1 \mid x_t) \approx q_t^\varphi(x_0, x_1 \mid x_t), \tag{19}$$

and train it by maximizing the joint endpoint likelihood,

$$\mathcal{L}_{\text{2S-VFM}}(\varphi) := -\mathbb{E}_{t,x_0,x_1}[\log q_t^\varphi(x_0, x_1 \mid x_t)] \quad \text{where } x_t = \alpha_t x_0 + \beta_t x_1. \tag{20}$$

Under a mean-field endpoint factorization, the objective decomposes into standard supervised terms,

$$\mathcal{L}_{\text{2S-VFM}}(\varphi) = -\mathbb{E}_{t,x_0,x_1}\left[\sum_{d=1}^{D}\left(\log q_t^\varphi(x_0^d \mid x_t) + \log q_t^\varphi(x_1^d \mid x_t)\right)\right]. \tag{21}$$

This recovers familiar per-component losses (Gaussian likelihoods for continuous variables; cross-entropy for categorical variables) while treating both endpoints symmetrically.

Let the posterior means (i.e. predicted endpoints) be

$$\mu_{0,t}^\varphi(x_t) := \mathbb{E}_{q_t^\varphi(x_0|x_t)}[x_0], \qquad \mu_{1,t}^\varphi(x_t) := \mathbb{E}_{q_t^\varphi(x_1|x_t)}[x_1]. \tag{22}$$

Plugging these into equation 13 yields a stable induced velocity estimator,

$$v_t^\varphi(x_t) = \dot{\alpha}_t\,\mu_{0,t}^\varphi(x_t) + \dot{\beta}_t\,\mu_{1,t}^\varphi(x_t), \tag{23}$$

which avoids ill-conditioned divisions and remains well-behaved over $t \in [0, 1]$.

Moreover, this two-sided form provides a direct handle for structured supports: if we parameterize $q_t^\varphi(x_1 \mid x_t)$ such that $\mu_{1,t}^\varphi(x_t) \in \Omega$ by construction, then the learned velocity inherits endpoint-induced affine confinement in practice via equation 18. Empirically, we observe this formulation to provide more stable training dynamics (see Appendix C), and thus we adopt the two-sided formulation for all our experiments.

## 3.3 Pawsterior: conditional two-sided VFM for SBI

Flow-matching posterior estimation (FMPE) (Wildberger et al., 2023) learns a conditional velocity field by regressing onto endpoint differences. Given simulator pairs $(\theta_1, x) \sim \mathcal{D}$, a base draw $\theta_0 \sim p_0$, and $t \sim \mathcal{U}[0, 1]$, it minimizes

$$\mathcal{L}_{\text{FMPE}}(\varphi) = \mathbb{E}_{(\theta_1,x),\,\theta_0,\,t}\left[\|v_t^\varphi(\theta_t; x) - (\theta_1 - \theta_0)\|_2^2\right], \qquad \theta_t = (1 - t)\theta_0 + t\theta_1. \tag{24}$$

Under common parameterizations this corresponds to Gaussian mean-matching over endpoints, so feasibility constraints are not explicitly enforced and capacity may be spent on invalid directions.

To incorporate the confinement principle from §3.1, we adopt the stable two-sided endpoint parameterization from §3.2. Instead of learning $v_t^\varphi$ directly, we learn conditional endpoint posteriors given an intermediate state $\theta_t$ and observation context $x$:

$$p_t(\theta_0, \theta_1 \mid \theta_t, x) \approx q_t^\varphi(\theta_0, \theta_1 \mid \theta_t, x), \qquad \theta_t = \alpha_t \theta_0 + \beta_t \theta_1. \tag{25}$$

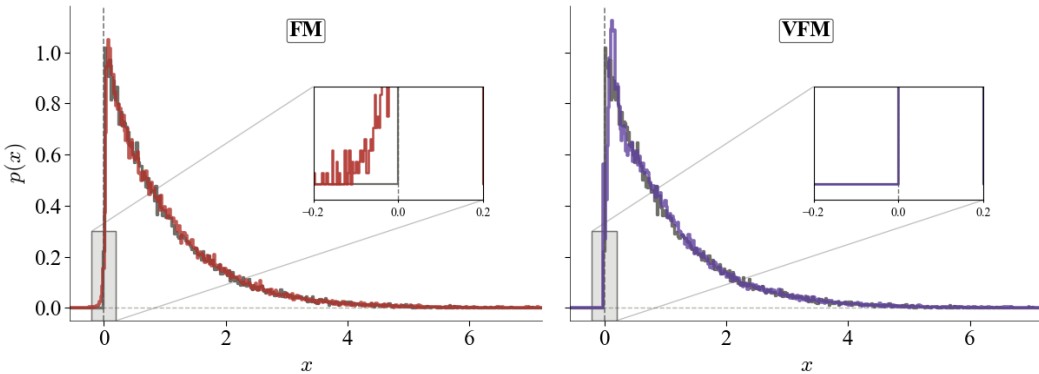

Figure 1: Toy example illustrating the effect of geometric constraints. Standard FM transports probability mass through infeasible regions of the parameter space, whereas the proposed endpoint-based variational formulation restricts the flow to the feasible set.

Following Eijkelboom et al. (2025), we can define a *conditional* VFM objective for this setting. Assuming a mean-field objective and letting the endpoint means be given by

$$\mu_{0,t}^{\varphi}(\theta_t, x) := \mathbb{E}_{q_t^{\varphi}(\theta_0|\theta_t,x)}[\theta_0], \qquad \mu_{1,t}^{\varphi}(\theta_t, x) := \mathbb{E}_{q_t^{\varphi}(\theta_1|\theta_t,x)}[\theta_1], \qquad (26)$$

which would induce the conditional velocity via

$$v_t^{\varphi}(\theta_t; x) = \dot{\alpha}_t \, \mu_{0,t}^{\varphi}(\theta_t, x) + \dot{\beta}_t \, \mu_{1,t}^{\varphi}(\theta_t, x). \qquad (27)$$

If $q_t^{\varphi}(\theta_1 \mid \theta_t, x)$ is any distribution such that $\mu_{1,t}^{\varphi}(\theta_t, x) \in \Omega$ by construction, the induced velocity is confined to the affine image $a_t \theta_t + c_t \Omega$, concentrating transport on feasible directions.

Training proceeds by maximizing the joint conditional endpoint likelihood,

$$\mathcal{L}_{\text{Pawsterior}}(\varphi) = -\mathbb{E}_{t,\theta_1,\theta_0,x} \left[ \sum_{d=1}^{D} \log q_t^{\varphi}(\theta_0^d \mid \theta_t, x) + \log q_t^{\varphi}(\theta_1^d \mid \theta_t, x) \right], \qquad (28)$$

where $\theta_0 \sim \mathcal{N}(0, \mathbf{I})$, $\theta_1 \sim p(\theta)$, $x \sim \text{Simulator}(\theta_1)$, $t \sim \mathcal{U}(0,1)$, and $\theta_t := t\theta_1 + (1 - t)\theta_0$. As such, choosing the appropriate distribution and hence loss for $\theta_1$ — e.g. Gaussian likelihoods (MSE) for continuous targets, categorical cross-entropy for discrete data — enables one to unify mixed-type SBI within a single amortized model.

## 4 EXPERIMENTS

We evaluate our method to test the central claim that explicitly modeling endpoint geometry and support constraints improves posterior estimation in simulation-based inference. Experiments cover two complementary regimes: standard SBI benchmarks with continuous parameters, where we assess improvements in posterior fidelity and stability under geometric constraints, and a categorical switching-regime task designed to probe the method's ability to handle discrete and hybrid parameter spaces that challenge conventional FM approaches.

### 4.1 SBI BENCHMARK

We first evaluate on the *sbibm* benchmark (Lueckmann et al., 2021), a widely used suite of simulation-based inference tasks. Each task specifies a prior distribution $p(\theta)$ together with a simulator generating observations as $x \sim p(x \mid \theta)$, enabling construction of paired datasets $(\theta, x)$ for training amortized posterior estimators approximating $p(\theta \mid x)$.

The benchmark additionally provides high-quality reference posterior samples, allowing quantitative evaluation of posterior fidelity. Following standard practice, we report the classifier two-sample test (C2ST) metric, which measures how well generated posterior samples match the reference distribution. A classifier is trained to distinguish reference from generated samples; performance near chance level (0.5) indicates close agreement between the two distributions.

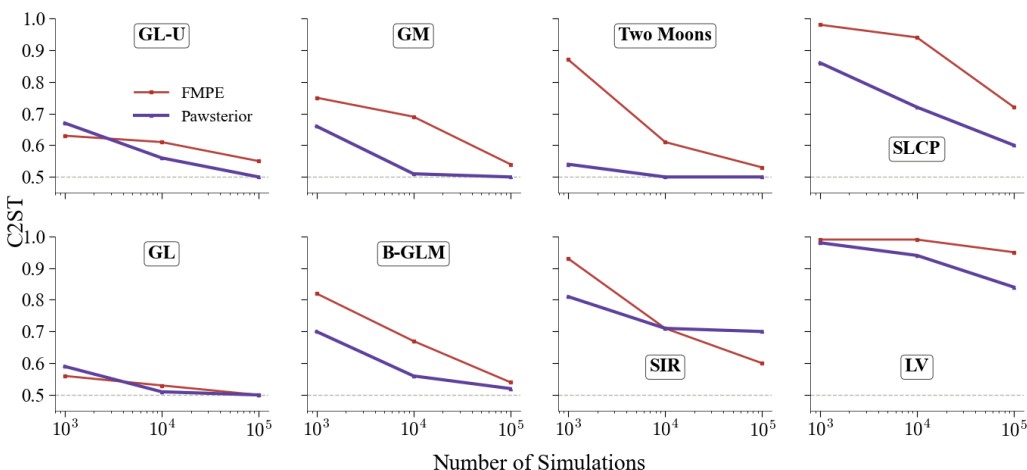

Figure 2: Comparison between FMPE and Pawsterior across *sbibm* tasks, evaluated using the C2ST metric. Lower values indicate better performance, with $0.5$ corresponding to samples that are indistinguishable from the reference posterior.

Figure 2 shows the performance of standard FMPE and the proposed Pawsterior across multiple *sbibm* tasks for models trained on $10^3, 10^4$ and $10^5$ simulations. The largest performance gains appear for the tasks shown in the upper half of the plot, whose posteriors have bounded support, where explicitly respecting the support structure leads to substantial improvements. Notably, however, Pawsterior also improves over FMPE on most tasks with unbounded posteriors, as shown in the lower half of the figure. This suggests that while support constraints amplify the benefits of our approach, the variational endpoint formulation can yield more stable and efficient learning even when no explicit bounds are present.

## 4.2 CATEGORICAL TASKS

To evaluate the proposed approach beyond standard continuous benchmarks, we introduce a synthetic task with purely *categorical* latent structure. Since the *sbibm* benchmark does not include discrete-parameter inference problems, this setting allows us to explicitly test whether our endpoint-based variational formulation can handle posteriors with discrete or hybrid support. We therefore design a custom simulation-based inference task based on a *Switching Gaussian Mixture* (SGM) model.

The SGM task consists of $K$ discrete regimes coupled to a continuous latent state evolving in $\mathbb{R}^{d_x}$. Let $T$ denote the number of transitions. A regime sequence $\theta = (z_0, \ldots, z_{T-1})$, with $z_t \in \{1, \ldots, K\}$, is first sampled from a Markov prior. Conditioned on this sequence, the continuous state evolves according to linear–Gaussian dynamics: the initial state $x_0$ is drawn from a Gaussian distribution, and each subsequent state $x_{t+1}$ is obtained by applying a regime-dependent linear transformation to $x_t$ followed by additive Gaussian noise. The observation therefore consists of the full trajectory $(x_0, \ldots, x_T)$, while the inference target is the discrete regime sequence $\theta$. Full details of the generative process are provided in Appendix A.

This task induces a posterior supported on a Cartesian product of categorical simplices, which violates the implicit continuous Euclidean assumptions underlying standard FMPE parameterizations. As a result, conventional FM approaches struggle to represent such posteriors faithfully. In contrast, Pawsterior accommodates categorical parameters by modeling endpoint distributions directly, allowing the learned transport to respect the discrete geometry of the posterior support.

Beyond correctness, this task also enables us to study the *parameter efficiency* of the two approaches. Because the posterior structure is simpler than in the *sbibm* benchmarks yet geometrically constrained, it provides a controlled setting to assess how architectural capacity interacts with support-aware inductive biases. We therefore deliberately consider smaller models and systematically vary network capacity. Specifically, we evaluate C2ST performance as a function of the number of resid-

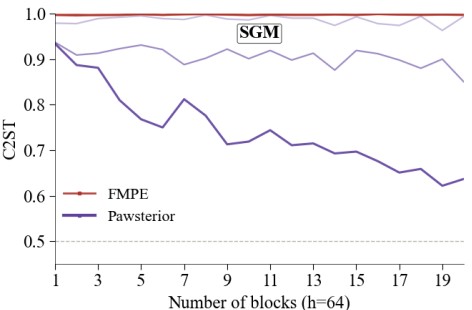 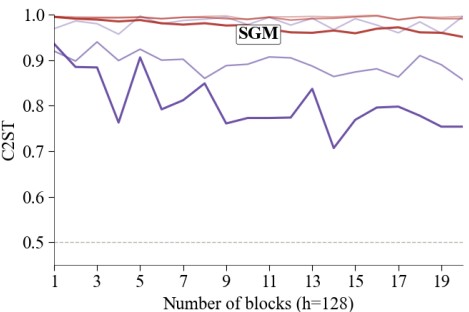

Figure 3: C2ST performance on the SGM task as a function of model depth (number of residual blocks) for hidden dimensions $h = 64$ (left) and $h = 128$ (right), across data regimes of $10^3$, $10^4$, and $10^5$ simulations. Lower-opacity curves correspond to fewer simulations, while higher-opacity curves indicate larger datasets.

ual blocks in a ResNet-style backbone (ranging from 1 to 20), for two hidden dimensions ($h = 64$ and $h = 128$), and across three data regimes ($10^3$, $10^4$, and $10^5$ simulations).

As shown in Figure 3, the performance gap between FMPE and Pawsterior is substantial. FMPE consistently struggles to capture the categorical posterior, yielding C2ST values close to $1.0$ even with up to $10^5$ training simulations. In contrast, Pawsterior improves steadily with increasing data and model capacity, reaching C2ST values around $0.6$. These results suggest that explicitly accounting for the discrete geometry of the posterior is critical: without an appropriate inductive bias, additional data alone does not suffice to recover meaningful inference.

## 5 CONCLUSION

In this work, we addressed a fundamental mismatch between standard flow-matching posterior estimation and the structured nature of simulation-based inference problems. While classical FM approaches typically assume unconstrained Euclidean parameter spaces, many SBI posteriors are supported on domains shaped by physical bounds, geometric constraints, or discrete structure. Ignoring this structure leads to inefficient transport and, in some cases, failure to recover meaningful posterior distributions. To resolve this, we introduced Pawsterior, a variational flow-matching framework that shifts the modeling perspective from unconstrained velocity regression to endpoint-aware variational inference. By explicitly modeling conditional endpoint distributions, Pawsterior inherits endpoint-induced affine geometric confinement, ensuring that transport trajectories remain aligned with the feasible set throughout the flow. This yields numerically stable velocity fields, naturally accommodates bounded or constrained domains, and enables coherent inference over discrete and mixed-type parameter settings.

Empirically, these geometric inductive biases translate into consistent performance gains. Pawsterior improves posterior fidelity across *sbibm* benchmarks, particularly for bounded-support posteriors, and succeeds on the SGM task where FMPE fails entirely, highlighting the importance of respecting posterior geometry in discrete settings. Moreover, Pawsterior demonstrates improved parameter efficiency in both data- and capacity-scaling regimes. Together, these findings suggest a broader principle: flow-matching methods benefit substantially when their inductive biases reflect the geometry and support structure of the inference problem, and explicitly modeling endpoint structure provides a scalable route to inference in constrained, discrete, and hybrid domains.

Looking forward, a key research direction is to more systematically understand how the geometry of the sample space shapes learned flows. This includes studying flows on bounded domains, simplices, and hybrid discrete–continuous spaces, as well as clarifying how endpoint-based parameterizations interact with these geometries during training and sampling. Such insights could inform principled architectural design for geometry-aware generative models and extend FM approaches to a wider class of structured inference problems where respecting support constraints is not optional but essential.

ACKNOWLEDGMENTS

JWvdM acknowledges support from the European Union Horizon Framework Programme (Grant Agreement No. 101120237). This project was supported by the Bosch Center for Artificial Intelligence.

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

## A  SWITCHING GAUSSIAN MIXTURE TASK

We define a switching linear–Gaussian state–space model with $K$ discrete regimes and continuous latent states in $\mathbb{R}^{d_x}$. Let $T$ denote the number of transitions, so that the discrete regime sequence has length $T$ and the continuous trajectory has length $T + 1$.

**Parameters and observations.**  The parameter of interest for simulation–based inference is the discrete regime path

$$\theta := z_{0:T-1} \in \{1, \ldots, K\}^T,$$

which is represented internally as a concatenation of one–hot vectors in $\{0, 1\}^{TK}$. The observation is the full continuous trajectory

$$x_{0:T} := (x_0, \ldots, x_T), \qquad x_t \in \mathbb{R}^{d_x},$$

which is flattened into a vector in $\mathbb{R}^{(T+1)d_x}$.

**Prior over regime sequences.**  The regime sequence $(z_t)_{t=0}^{T-1}$ follows a first–order Markov chain. The initial distribution is uniform,

$$z_0 \sim \text{Categorical}(\pi_0), \qquad \pi_0 = \tfrac{1}{K}\mathbf{1},$$

and transitions are governed by a *sticky* transition matrix $\Pi \in \Delta^{K \times K}$ of the form

$$\Pi = 0.3 \cdot \tfrac{1}{K}\mathbf{1}\mathbf{1}^\top + 0.7 \cdot I_K,$$

so that with high probability the process remains in the same regime.

**Regime–specific dynamics.** For each regime $k \in \{1, \dots, K\}$, we define:

- A linear dynamics matrix $A_k \in \mathbb{R}^{d_x \times d_x}$, constructed as a scaled rotation

$$A_k = 0.8\, R_k, \qquad R_k \in SO(d_x),$$

where $R_k$ is a random rotation matrix. The scaling factor ensures strict stability of the dynamics.

- An observation noise scale $\sigma_k > 0$, with values linearly spaced in the interval $[0.25, 0.6]$.

- A drift vector $b_k \in \mathbb{R}^{d_x}$, sampled as $b_k \sim \mathcal{N}(0, 2^2 I)$

**Initial state.** The initial continuous state is drawn from an anisotropic Gaussian distribution,

$$x_0 \sim \mathcal{N}(0, \Sigma_0), \qquad \Sigma_0 = \mathrm{diag}(s_0^2),$$

where the entries of $s_0 \in \mathbb{R}^{d_x}$ are linearly spaced between $0.3$ and $2.0$.

**Transition model.** Conditioned on the regime sequence, the continuous dynamics evolve according to

$$x_{t+1} = A_{z_t} x_t + b_{z_t} + \sigma_{z_t} \varepsilon_t, \qquad \varepsilon_t \sim \mathcal{N}(0, I),$$

for $t = 0, \dots, T - 1$.

**Likelihood factorization.** Given a regime path $z_{0:T-1}$, the likelihood of the observed trajectory factorizes as

$$p(x_{0:T} \mid z_{0:T-1}) = p(x_0) \prod_{t=0}^{T-1} p(x_{t+1} \mid x_t, z_t),$$

where

$$p(x_0) = \mathcal{N}(0, \Sigma_0), \qquad p(x_{t+1} \mid x_t, z_t = k) = \mathcal{N}(x_{t+1}; A_k x_t + b_k, \sigma_k^2 I).$$

The joint distribution of regimes and observations is therefore

$$p(z_{0:T-1}, x_{0:T}) = p(z_0) \prod_{t=1}^{T-1} p(z_t \mid z_{t-1}) \cdot p(x_0) \prod_{t=0}^{T-1} p(x_{t+1} \mid x_t, z_t).$$

**Posterior sampling.** The SBI target posterior is

$$p(\theta \mid x_{0:T}) = p(z_{0:T-1} \mid x_{0:T}).$$

Since the latent variables $z_t$ are discrete and the emission likelihood at time $t$ depends only on $(x_t, x_{t+1})$, the posterior can be sampled exactly using forward–filtering backward–sampling (FFBS).

We define the per–time log–likelihoods

$$\ell_t(k) := \log p(x_{t+1} \mid x_t, z_t = k) = -\tfrac{1}{2} \left( \frac{\|x_{t+1} - (A_k x_t + b_k)\|^2}{\sigma_k^2} + d_x \log(2\pi \sigma_k^2) \right).$$

**Forward pass.** Let $\alpha_t(k) \propto p(z_t = k \mid x_{0:T})$. In log space,

$$\log \alpha_0(k) \propto \log \pi_0(k) + \ell_0(k),$$

and for $t \geq 1$,

$$\log \alpha_t(k) \propto \ell_t(k) + \log \sum_{j=1}^{K} \exp(\log \alpha_{t-1}(j) + \log \Pi_{j,k}).$$

At each time step, the forward messages are normalized using a log–sum–exp operation.

**Backward sampling.** We first sample

$$z_{T-1} \sim \mathrm{Categorical}(\alpha_{T-1}),$$

and then for $t = T - 2, \dots, 0$,

$$p(z_t = j \mid z_{t+1} = k, x_{0:T}) \propto \alpha_t(j)\, \Pi_{j,k}.$$

This procedure yields exact samples from the posterior $p(z_{0:T-1} \mid x_{0:T})$, which are then converted to their one–hot parameterization $\theta$.

## B   EXPERIMENTAL SETUP

To ensure a fair comparison, we follow the experimental protocol of Wildberger et al. (2023) as closely as possible for the *sbibm* experiments. We use a residual MLP (ResNet) architecture to parameterize the endpoint predictors. For each task, we run a subset of the hyperparameter grid described in Table 1. We use a fixed batch size of 1024.

**Optimization**   We optimize all models using Adam with the learning rate selected from the corresponding grid. We apply gradient clipping with maximum norm 1.0. A ReduceLROnPlateau scheduler is used on the validation loss with factor 0.5 and patience 50 epochs.

**Hardware**   All experiments are run on NVIDIA A100 GPUs. We use automatic mixed precision (AMP) during training; on Ampere-class GPUs this corresponds to bfloat16 autocasting.

**Constrained supports**   For bounded continuous parameters, we enforce support constraints by mapping unconstrained network outputs to the interval $[\text{low}, \text{high}]$ using a $\tanh$ squashing transformation. For categorical blocks, training is performed in the unconstrained logit space, while at sampling time we project to the corresponding simplex performing softmax operations and the final samples are projected to hard one-hot vectors using an argmax operation after the final integration step.

**Sampling**   Posterior samples are generated by Euler integration of the learned flow using 100 steps.

**Time prior sampling (*sbibm*)**   A uniform time prior $t \sim \mathcal{U}[0, 1]$ distributes training capacity evenly across the interpolation path. In practice, however, the complexity of the vector field can vary substantially with $t$, and for bounded or sharply constrained targets the most challenging region often occurs near $t \approx 1$. For the *sbibm* experiments, FMPE and Pawsterior therefore optionally sample $t$ from a power-law distribution

$$t = u^{\frac{1}{1+\alpha}}, \qquad u \sim \mathcal{U}[0, 1],$$

which induces a density $p_\alpha(t) \propto t^\alpha$ on $[0, 1]$. This recovers the uniform prior at $\alpha = 0$ and increasingly emphasizes late times for $\alpha > 0$. This heuristic has been shown to improve learning in settings with sharp bounds by allocating more capacity to the near-target part of the transport.

**SGM-specific setup**   For the SGM experiment, we intentionally study smaller architectures to analyze parameter efficiency and scaling behavior. In this setting, we fix the time prior to the uniform distribution ($\alpha = 0$) and do not perform a sweep over $\alpha$. We consider two hidden dimension regimes, $h \in \{64, 128\}$. For each hidden dimension, we sweep the learning rate over $\{10^{-3}, 10^{-4}\}$ and vary the number of residual blocks from 1 to 20. Performance is evaluated using the C2ST metric across different data regimes. As for the specific task configuration, we consider $T = 10$ timesteps, $K = 10$ categorical regimes, and $d_x = 5$ dimensionality of the observations.

Table 1 summarizes the hyperparameter ranges considered in the *sbibm* experiments.

Table 1: Hyperparameter grid used for the *sbibm* experiments.

| Hyperparameter | Values |
|---|---|
| Hidden dimension | $2^n$ for $n \in \{9, 10\}$ |
| Number of residual blocks | $\{15, 16, 17, 18\}$ |
| Learning rate | $\{10^{-3}, 10^{-4}\}$ |
| Time prior exponent $\alpha$ | $\{-0.5, -0.25, 0, 1, 4\}$ |
| Batch size | 1024 (fixed) |

## C  1-SIDED RESULTS

Empirically, we observed that two-sided prediction yields more stable training dynamics. This became evident in the early stages of the project during evaluation on sbimb tasks, so for computational constraints we evaluated only using the two-sided formulation. The corresponding results for the SGM task are shown in Figure 4.

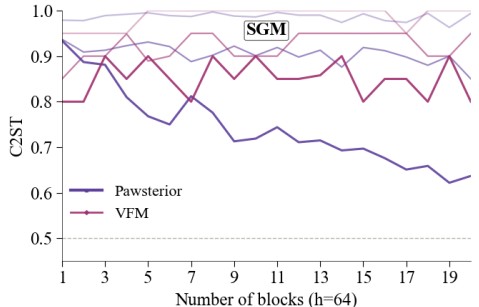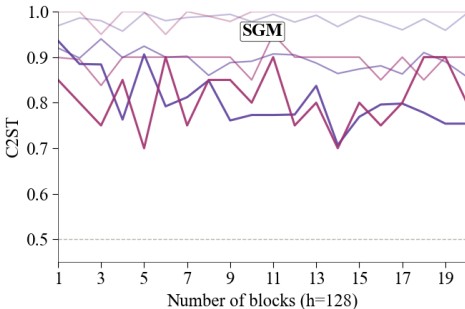

Figure 4: Comparison of C2ST performance on the SGM task as a function of model depth (number of residual blocks) for hidden dimensions $h = 64$ (left) and $h = 128$ (right), between 1-sided endpoint prediction (VFM) and 2-sided (Pawsterior).

