# OpenReview forum: "Pawsterior: Variational Flow Matching for Structured Simulation-Based Inference"
_ICLR.cc/2026/Workshop/GRaM — ICLR 2026 Workshop GRaM Poster_

### Official Review · Reviewer_rRAo · 2026-02-18
**This paper proposes Pawsterior, a conditional two-sided variational flow matching approach for simulation-based inference (SBI) that aims to better respect structured parameter supports (bounded domains, simplices, and hybrid discrete–continuous settings). The core idea is to shift from direct Euclidean velocity regression to endpoint distribution modeling, leveraging the fact that the population flow-matching target already encodes endpoint geometry via conditional expectations. The authors formalize an “endpoint-induced affine geometric confinement” principle and argue that two-sided endpoint prediction yields improved numerical stability by avoiding ill-conditioned divisions near the boundaries of the time interval. Empirically, they report improved C2ST on sbibm tasks (especially bounded-support ones) and a synthetic switching-regime task where the baseline FMPE fails to capture a categorical posterior.**

**Rating:** 6
**Confidence:** 5

**Review:**

Overall assessment
The paper addresses a real limitation of standard flow-matching posterior estimators for SBI: mismatch between unconstrained Euclidean vector fields and structured posterior supports. The proposed endpoint-based formulation is conceptually well-motivated and the two-sided endpoint prediction is a practical stabilization that is easy to adopt in existing FMPE codebases. The empirical results suggest meaningful gains on bounded-support sbibm tasks and a clear advantage in a categorical regime-switching toy problem.
At the same time, the paper would be significantly stronger with (i) more comprehensive baselines for discrete/hybrid inference, (ii) more diagnostic SBI evaluations beyond C2ST, and (iii) clearer discussion of when the mean-field endpoint factorization is sufficient versus when dependencies between discrete and continuous components matter.

Key technical contributions (as I understand them)
1.	Endpoint-induced affine geometric confinement. For affine interpolation
[
x_t = \alpha_t x_0 + \beta_t x_1,
]
the population target velocity can be written as
[
u_t(x_t)=a_t x_t + c_t,\mathbb{E}[x_1\mid x_t],\quad
a_t=\frac{\dot\alpha_t}{\alpha_t},;; c_t=\dot\beta_t-a_t\beta_t,
]
and if (x_1\in\Omega) with (\Omega) convex, then (\mathbb{E}[x_1\mid x_t]\in\Omega), implying a confinement set inclusion
[
u_t(x_t)\in a_t x_t + c_t \Omega.
]
2.	Two-sided endpoint prediction for stability. Instead of recovering one endpoint through division (ill-conditioned near (t\approx 0) or (t\approx 1)), the method predicts both endpoint means and uses
[
v_\phi(t,x_t)=\dot\alpha_t,\mu_{\phi,0,t}(x_t)+\dot\beta_t,\mu_{\phi,1,t}(x_t),
]
which is numerically stable across (t\in[0,1]).
3.	Conditional SBI version (Pawsterior). Learn
[
p_t(\theta_0,\theta_1\mid \theta_t,x)\approx q_{\phi,t}(\theta_0,\theta_1\mid \theta_t,x),
]
and compute the conditional velocity analogously from the predicted endpoint means, with support constraints enforced by construction (for bounded continuous variables) and simplex projections (for categorical blocks).
________________________________________
Strengths / Pros
•	Clear motivation and good alignment with known limitations of Euclidean FM in SBI. The paper articulates why “invalid-region transport” is not just inefficient but can be fundamentally incompatible with discrete supports.
•	Conceptually simple and implementable. The shift from velocity regression to endpoint likelihood objectives (per-coordinate Gaussian / cross-entropy losses) is straightforward, and should be relatively easy to plug into existing FMPE pipelines.
•	Two-sided stabilization is practically valuable. Avoiding divisions by (\alpha_t) or (1-t) is a convincing engineering improvement for training and sampling stability.
•	Empirical signal on standard SBI benchmarks. The sbibm C2ST comparisons indicate consistent gains, especially where bounded supports matter.
•	Discrete task stress test is a useful addition. The SGM regime-switching experiment highlights a case where conventional FMPE parameterizations struggle severely, supporting the claim that endpoint distributions are the right abstraction for categorical/hybrid SBI.
________________________________________
Weaknesses / Cons
•	Baseline coverage is limited for the discrete/hybrid claim. The categorical experiment compares primarily against FMPE, which is plausibly a weak or mismatched baseline for one-hot / simplex-supported posteriors. The paper would benefit from comparisons to methods that natively handle categorical or simplex geometry (even if only on the synthetic task), or at least to a CatFlow-style endpoint model without the two-sided SBI-specific modifications.
•	Evaluation relies heavily on C2ST. C2ST is useful but not sufficient for SBI. For SBI, additional metrics such as simulation-based calibration (SBC), posterior coverage / rank statistics, posterior predictive checks, and task-specific accuracy (for the switching path) would strengthen the empirical story substantially.
•	Mean-field endpoint factorization may be restrictive. The objective decompositions suggest a per-component loss, but discrete–continuous dependencies (or correlated continuous parameters) can be critical in SBI. It is not yet clear when the mean-field approximation is harmless versus when it loses important posterior structure.
•	Discrete sampling via projection and argmax needs more justification. The approach uses softmax projection to the simplex and a final argmax after ODE integration. This is pragmatic, but it introduces a non-smooth “rounding” step that can affect correctness, calibration, and theoretical interpretation of the learned flow. More discussion and ablations would help.
•	Support enforcement via squashing may distort geometry. Mapping unconstrained outputs to ([{\rm low},{\rm high}]) through (\tanh) is standard, but it can bias densities near boundaries. A discussion of potential distortion and whether alternatives (e.g., logit transforms for bounded intervals) change results would be valuable.
________________________________________
Questions for the authors
1.	Discrete task: why is C2ST still far from 0.5? The categorical SGM experiment reports improvement but remains around (\sim 0.6). Is the residual gap due to model capacity, the projection/argmax procedure, or the training objective?
2.	Ablation: how much gain is from two-sided endpoints versus constraint enforcement? A clean ablation (one-sided vs two-sided; with vs without constraint mapping; fixed time prior vs power-law) would clarify which component drives improvements.
3.	Mean-field adequacy: Have you observed failure cases where dependencies across coordinates are important? Would a structured (q_{\phi,t}(\theta_1\mid \theta_t,x)) (e.g., low-rank Gaussian, coupling layers, autoregressive categorical blocks) improve performance?
4.	Fairness of hyperparameter budgets: Are FMPE and Pawsterior tuned with equal compute and search space (learning rate, time prior exponent, architecture depth), and are the reported comparisons the best-per-method?

Suggestions for improvement
•	Add SBI-calibration metrics (SBC ranks, coverage curves) and at least one posterior predictive check on sbibm tasks.
•	For the SGM task, add task-native metrics such as per-time-step regime accuracy, Hamming distance on (z_{0:T-1}), or KL between true and estimated marginal regime posteriors (where exact FFBS marginals are available).
•	Include stronger discrete/hybrid baselines (or clearly justify why they are not feasible) and provide an ablation isolating the contribution of two-sided endpoint prediction.
•	Clarify the theoretical scope of the confinement argument beyond convex (\Omega), and discuss what happens for non-convex feasible sets (common in real simulators).

**Pmlr Suitability:**

Yes

---

### Official Review · Reviewer_sR5Y · 2026-02-24
**Potentially intresting idea with limited empirical evidence**

**Rating:** 3
**Confidence:** 5

**Review:**

The paper introduces a parameter constraint-aware method for simulation-based posterior estimation (Simulation Based Inference - SBI). The method uses the idea of Variational Flow Matching (VMF) of learning endpoint distributions and uses it as an amortized conditional sampler for $p(\theta\mid x)$ with constraint-handling essentially baked in.

The authors argue that conventional flow matching approaches treat parameter spaces as unconstrained Euclidean, and therefore spend computational resources during simulation based inference in computations with parameters outside of the feasible set relevant for the problem at hand.
Instead, in this manuscript, the authors target problems where posteriors exhibit structured
support imposed by physical constraints, bounds, symmetries, or discrete latent structure, and in order to accommodate such problems they propose a **two-sided variational formulation of flow matching**, inspired by the one-sided variational formulation of the approach ( Eijkelboom et al. (2024) ). The method trains a neural network to predicts both endpoints (noise endpoint from initial distribution and data endpoint from the target) of the bridge that connects the initial and target distribution from intermediate points. The authors claim that this is necessary to avoid numerical instabilities from one-sided formulas (used in VMF).
They report that this “two-sided” endpoint formulation results in posterior samples that are harder for a classifier to distinguish from reference samples on sbibm tasks (lower classifier two-sample
test (C2ST) metric, interpreted as better posterior fidelity), and they also apply the approach on a toy example where the unknown parameters include discrete variables (a switching Gaussian mixture)

---
## Strengths

- The paper reformulates flow-matching SBI as conditional endpoint prediction $(\theta_0,\theta_1)\mid(\theta_t,x)$, providing thus a simple supervised learning formulation that fits amortized SBI quite well.
- By modeling endpoints with appropriate likelihoods (Gaussian for continuous, categorical for discrete), the proposed approach offers a unified framework for dealing with hybrid continuous–discrete parameter posteriors in SBI.


---

## Weaknesses

- Limited empirical support of the claims.
- As I understand, plots correspond to single experiments/no statistics over independent runs.
- The are no ablations presented, especially ablations on the two-endpoint/single-endpoint conditioning that is the major contribution compared to VFM. The paper lacks ablations that help understand and identify whether claimed improvements come from two-sided endpoints, constraint parameterization, or time-weighting choices.
- No baseline comparisons with other methods that consider constraints in Flow Matching frameworks.
- The "geometric confinement" result is a consequence of basic convexity/conditional expectation.
- The discrete/categorical claim is supported only on a toy switching-mixture experiment evaluated with the C2ST metric, without stronger task-appropriate metrics or comparison with discrete SBI baselines.

---
## Major issues


I find the approach interesting but I expect for a full length paper published in PMLR a more thorough numerical
exploration than what the authors provide in the current version of the manuscript. In my opinion the numerical support and evidence of the method best suits a short/workshop paper as opposed to a full length archival manuscript.

 I would expect the authors to provide:
1. comparisons with existing approaches of introducing constraints in SBI/flow matching problems.  For example, some bounded domains can often be accounted for with reparametrisation (e.g. a sigmoid for box constraints or soft plus for positivity).

2. comparisons with flow matching approaches directly on the manifold (geometry-aware probability paths).
   Existing flow matching frameworks like Fisher Flow Matching [1], which treats the probability simplex as a Riemannian manifold (Fisher–Rao metric) and Stiefel flow matching [2], solve flow matching problems consistent with the considered geometry. How would that compare with the considered two0seded variational approach?

3. comparisons with methods that intriduce constraints during sampling by projecting, reflecting or otherwise correcting the some pretrained flow or boundary/constraint aware sampling/stochastic processes.  For example, physics-/constraint-corrected flow matching [4], Gauge flow matching [5], reflected diffusion models [6] or diffusion models for constrained domains [7] , mirror-map approaches [8], geodesic approaches [3, 9] and many more [10].





Thus I would also expect the numerical experiments to reflect the different variants of constraints the method could account for and compare with the fitting baseline(s).
For instance I would start from simple boundary/positivity constraints, and compare with a reparametrizing approach
Then consider some sort of nontrivial geometry (manifolds, simplifies) and compare with associated baselines
Consider arbitrary constraints and compare with some flavor of Gauge flow matching etc.

These experiments would clarify which sorts of constraints the propose approach can accommodate and would clarify more the contribution of the paper.

- Moreover I would expect a clear positioning of the present work in terms of these geometry or boundary/constrained aware frameworks. Thus I would suggest an extensive related works section in the future iteration of the manuscript.

- See also the weaknesses

---
## Minor issues

- The claim/framing of framing “previously inaccessible” structured SBI problems is overstated based on the discussion above and  relative to what is shown in the current version of the manuscript.
- The core theoretical part of the paper mainly re-derives the (Variational) Flow Matching approach and then proposes the two sided conditioning, which reads more like a recipe as opposed to a new framework.

- In Eq. 16 please avoid using alpha and a together in same set of equations, since it is difficult to discern which one is which. Please use other letters.

---
## Summary

The paper introduces an interesting variant of flow matching that accommodates constrained parameter spaces. However, as is the paper reads more like an engineering variant of conditional variational flow matching, than a concrete new framework. The limited numerical validation with lack of proper statistical considerations, ablations and comparisons to existing baselines do not help to convince the reader of the importance of the proposed approach. The claims of “previously inaccessible” sampling approach, and “stable and consistently better” method need stronger empirical validation.
Moreover, the paper warrants a section that positions the approach in the field of similar approaches mentioned above either for constrained diffusion models or constrained flow matching.


---
## References

[1] Davis, O., Kessler, S., Petrache, M., Ceylan, İ. İ., Bronstein, M., & Bose, A. J. (2024). Fisher flow matching for generative modeling over discrete data. Advances in Neural Information Processing Systems, 37, 139054-139084.

[2] Cheng, A., Lo, A., Lee, K. L. K., Miret, S., & Aspuru-Guzik, A. (2024). Stiefel flow matching for moment-constrained structure elucidation. arXiv preprint arXiv:2412.12540.

[3] Kapusniak, K., Potaptchik, P., Reu, T., Zhang, L., Tong, A., Bronstein, M., ... & Di Giovanni, F. (2024). Metric flow matching for smooth interpolations on the data manifold. Advances in Neural Information Processing Systems, 37, 135011-135042.

[4] Utkarsh, U., Cai, P., Edelman, A., Gomez-Bombarelli, R., & Rackauckas, C. V. (2025). Physics-constrained flow matching: Sampling generative models with hard constraints. arXiv preprint arXiv:2506.04171

[5] Li, X., Liang, E., & Chen, M. Gauge Flow Matching: Efficient Constrained Generative Modeling over General Convex Set and Beyond. In The Fourteenth International Conference on Learning Representations.

[6] Lou, A., & Ermon, S. (2023, July). Reflected diffusion models. In International Conference on Machine Learning (pp. 22675-22701). PMLR.

[7] Fishman, N., Klarner, L., De Bortoli, V., Mathieu, E., & Hutchinson, M. (2023). Diffusion models for constrained domains. arXiv preprint arXiv:2304.05364.

[8] Liu, G. H., Chen, T., Theodorou, E., & Tao, M. (2023). Mirror diffusion models for constrained and watermarked generation. Advances in Neural Information Processing Systems, 36, 42898-42917

[9] Maoutsa, D. (2025). From geometry to dynamics: Learning overdamped Langevin dynamics from sparse observations with geometric constraints. arXiv preprint arXiv:2512.23566.

[10] Nordenhög, A., & Sharma, A. (2025). Score-based constrained generative modeling via Langevin diffusions with boundary conditions. arXiv preprint arXiv:2510.23985.

**Pmlr Suitability:**

No

---

### Meta-Review · Area_Chair_nd9P · 2026-02-26

**Decision:**

Accept

**Metareview:**

The paper introduces a novel approach for simulation-based inference. The reviewers mention they find the approach interesting with clear motivation. The main weakness is that the paper lacks a thorough empirical evaluation, making it hard to identify exactly where the claimed improvements come from and more/better baselines are needed.
However, the paper does contain a novel idea with some empirical results and no issues with the technical soundness are mentioned. Therefore, following the GRaM guidelines, I recommend this paper to be accepted, but would urge the authors to consider the feedback from the reviewers in improving the paper.

**Relevance To Proceedings:**

Yes — suitable for PMLR (long paper)

**Relevance To Workshop:**

Yes — suitable for GRaM

---

### Decision · Program_Chairs · 2026-03-02

Accept (Poster)